# EFFICIENT OFFLINE PREFERENCE-BASED REINFORCEMENT LEARNING WITH TRANSITION-DEPENDENT DISCOUNTING

## ABSTRACT

Offline preference-based reinforcement learning (OPBRL) tackles two major limitations of traditional reinforcement learning: the need for online interaction and the requirement for carefully designed reward labels. Despite recent progress, solving complex tasks with a small number of preference labels remains challenging, as the learned reward function is inaccurate when preference labels are scarce. To tackle this challenge, we first demonstrate that the inaccurate reward model predicts low-preference regions much more precisely than high-preference regions, as the former suffers less from generalization errors. By incorporating this insight with offline RL's pessimism property, we propose a novel OPBRL framework, Transition-dEpendent Discounting (TED), that excels in complex OPBRL tasks with only a small number of preference queries. TED assigns low transition-dependent discount factors to the predicted low-preference regions, which discourages the offline agent from visiting these regions and achieves higher performance. On the challenging Meta-World MT1 tasks, TED significantly outperforms current OPBRL baselines.

## 1 INTRODUCTION

Traditional Deep Reinforcement Learning (DRL) has demonstrated remarkable success in scenarios where online interactions with the environment are easy to acquire and accurate reward annotations are accessible (Silver et al., 2017b; Vinyals et al., 2019; Ceron & Castro, 2021). However, numerous real-world situations—such as those involving robotics—fail to meet these prerequisites, considerably curtailing the applicability of DRL. Offline preference-based reinforcement learning (OPBRL) addresses this issue by learning from a fixed unlabelled offline dataset and querying experts for their preference labels over pairs of trajectory segmentations (Rafailov et al., 2023; Kang et al., 2023). This learning paradigm is widely applicable to a range of real-life domains, e.g., robots learning to perform personalized behaviors from past experiences and user preferences.

Previous works on OPBRL perform similarly to offline RL with ground-truth rewards on simple D4RL benchmarks (Shin et al., 2022; Kim et al., 2022). However, they require a large number of preference queries to solve complex tasks like Meta-World MT1 (Yu et al., 2020), as these methods need to learn a precise reward model from preference labels. So, in this study, we aim to answer the following question:

*How to achieve efficient OPBRL with a small number of preference queries?*

In real-life applications, offline datasets are usually of low quality and contain massive noise, as these datasets may contain sub-optimal human behaviors and exploratory data. What's more, both Li et al. (2023) and our empirical results reveal that offline RL is robust to reward quality if the dataset is of high quality, so we focus on the more realistic and challenging setting where the dataset has massive low-preference data. When the number of preference queries is small, high-preference regions are hard to predict precisely by the learned reward model, as the data distribution for these data is usually narrow, and the predictions suffer significantly from generalization errors. In contrast, the learned reward model can identify low-preference regions much more precisely, as the model is more likely to generalize low-preference predictions to the ground-truth low-preference regions.

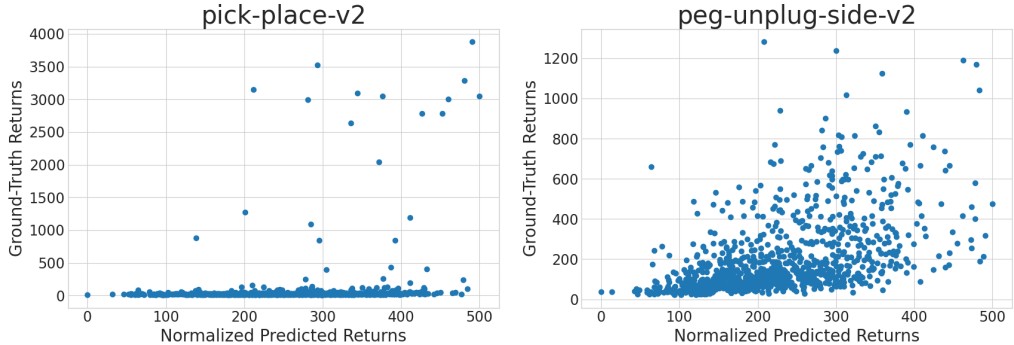

Figure 1: Visualization of normalized predicted returns and the ground-truth returns on two Meta-World tasks. The reward model predicts low-preference regions precisely (the bottom-left part of the figures), but fails to precisely predict high-preference regions (the right part of the figures) due to generalization errors. The reward models are learned with OPRL (Shin et al., 2022) and use ten preference queries.

Table 1: Our proposed method of assigning lower discount to the predicted low-preference regions, TED, can greatly improve the performance of current state-of-the-art (SOTA) OPBRL algorithms, OPRL-I (OPRL with IQL as the offline algorithm) and PT (Kim et al., 2022), and even achieve true-reward-level performance with merely ten queries on peg-unplug-side-v2.

| Task | True Reward | OPRL-I | OPRL-I+TED | PT | PT+TED |
|---|---|---|---|---|---|
| pick-place-v2 | **0.53±0.16** | 0.01±0.00 | 0.14±0.01 | 0.01±0.00 | 0.44±0.08 |
| peg-unplug-side-v2 | **0.37±0.10** | 0.30±0.06 | **0.43±0.04** | 0.25±0.05 | **0.39±0.09** |
| Average Over 50 Tasks | **0.65±0.07** | 0.33±0.02 | 0.39±0.03 | 0.27±0.03 | 0.40±0.04 |

Figure 1 gives a straightforward demonstration of this phenomenon, where high-preference regions (the right part of the figures) have large prediction errors, but low-preference regions (the bottom-left part of the figures) can be predicted well.

To achieve query-efficient OPBRL, we incorporate this interesting feature with offline RL's pessimism property, discouraging the agent from visiting regions out of the dataset support. We propose a novel OPBRL method, Transition-dEpendent Discounting (TED), which is conceptually simple, generally applicable to a range of off-the-shelf OPBRL algorithms, and achieves query-efficient OPBRL by assigning more pessimism to the predicted low-preference regions. In practice, TED replaces the original discount factor with transition-dependent discount factors before performing offline RL. By assigning lower discount factors to the predicted low-preference regions, the agent is discouraged from visiting these regions, thus attaining higher performance, as shown in Table 1.

As for empirical evaluation, we create a complex evaluation environment based on Meta-World MT1 (Yu et al., 2020). Results demonstrate that TED significantly improves performance when rewards are not accurate and does not harm performance if the reward is accurate. We also find that most D4RL tasks are insensitive to reward quality, which suggests that it might not be a suitable benchmark for OPBRL. To summarize, our contributions are listed as follows:

1. We propose a simple and effective OPBRL method TED, which achieves query-efficient learning and is generally applicable to a series of OPBRL algorithms.

2. We empirically demonstrate TED's power to improve performance greatly with even an inaccurate reward model.

3. We propose a new challenging OPBRL benchmark based on Meta-World MT1, and demonstrate that D4RL may not serve as a proper OPBRL benchmark due to its insensitivity to reward quality.

## 2 PRELIMINARIES

Reinforcement learning (RL) deals with Markov Decision Processes (MDPs). A MDP can be modelled by a tuple $(\mathcal{S}, \mathcal{A}, r, p, \gamma)$, with state space $\mathcal{S}$, action space $\mathcal{A}$, reward function $r(s, a)$, transition function $p(s'|s, a)$, and discount factor $\gamma$(Sutton & Barto, 2018). We follow the common assumption that the reward function is positive and bounded (Strehl et al., 2006): $\forall s \in \mathcal{S}, a \in \mathcal{A}, 0 \leq r(s, a) \leq R_{max}$, where $R_{max}$ is the maximum possible reward. RL's objective is to find a policy $\pi(a|s)$ that maximizes the cumulative discounted return $R(\pi) = \mathbb{E}_\pi\left[\sum_{t=0}^\infty \gamma^t r(s_t, a_t)\right]$. The Q function of a policy $\pi$ is defined as:

$$Q_\pi(s, a) = r(s, a) + \gamma \mathbb{E}_{s' \sim p(\cdot|s,a), a' \sim \pi(\cdot|s')}[Q_\pi(s', a')]. \tag{1}$$

OPBRL learns a return-maximizing policy $\pi(a|s)$ from a fixed dataset $D$ that does not have reward labels: $D = \{(s_i, a_i, s_{i+1})\}_{i=1}^N$, where $N$ is the size of the dataset. We follow the common assumption that $D$ consists of multiple trajectories (Fu et al., 2020; Wang et al., 2023). OPBRL agents can choose $N_p$ pairs of $H$-length trajectory segmentations $(\sigma^0, \sigma^1) = (\{(s_i^0, a_i^0, s_{i+1}^0)\}_{i=1}^H, \{(s_i^1, a_i^1, s_i^1)\}_{i=1}^H)$ from $D$ and query an expert for its preference label $y \in \{0, 1, 0.5\}$ over these two segmentations. We use the notation $\sigma^i \succ \sigma^j$ to indicate $\sigma^i$ is more preferrable than $\sigma^j$. $y = 0$ indicates that $\sigma^0 \succ \sigma^1$, $y = 1$ indicates that $\sigma^1 \succ \sigma^0$, and $y = 0.5$ indicates that both segmentations are equally preferrable. All preference data is stored in a preference dataset $D_p = \{(\sigma_i^0, \sigma_i^1, y_i)\}_{i=1}^{N_p}$.

A popular OPBRL framework (Shin et al., 2022; Kim et al., 2022) is first to learn a reward model $\hat{r}$ from preference labels, then acquire a labeled dataset $D_{\hat{r}} = \{(s_i, a_i, \hat{r}(s_i, a_i), s_{i+1})\}_{i=1}^N$ by labeling $D$ with the reward model, and finally perform standard offline RL algorithms on $D_{\hat{r}}$. A current SOTA OPBRL algorithm OPRL (Shin et al., 2022) learns a reward model and predicts preferences based on the sum of the rewards using the Bradley-Terry model (Bradley & Terry, 1952):

$$P_\psi(\sigma^1 \succ \sigma^0) = \frac{\exp(\sum_t \hat{r}_\psi(s_t^1, a_t^1)}{\sum_{j \in \{0,1\}} \exp(\sum_t \hat{r}_\psi(s_t^j, a_t^j)}, \tag{2}$$

where $\hat{r}_\psi(s_t^1, a_t^1)$ is the predicted reward for $(s_t^1, a_t^1)$, and $\psi$ is the parameter of the reward model. OPRL trains the reward model by minimizing the cross entropy loss between predicted preferences and the ground-truth preference labels:

$$\mathcal{L}^{CE}(\psi) = - \mathbb{E}_{(\sigma^0, \sigma^1, y) \sim D_p}\left[(1-y)\log P_\psi(\sigma^0 \succ \sigma^1) + y \log P_\psi(\sigma^1 \succ \sigma^0)\right]. \tag{3}$$

Transition-dependent discounting (Sharma et al., 2021) is an extension of state-dependent discounting (Mahmood et al., 2015; Wei & Guo, 2011; Stachurski & Zhang, 2021). It models the discount factor as a function of the transition pairs$(s, a, s')$, which is denoted as $\hat{\gamma}(s, a, s')$, where $0 \leq \hat{\gamma}(s, a, s') \leq 1, \forall(s, a, s')$. Then, the Q function with the transition-dependent discount is:

$$Q_\pi^{\hat{\gamma}}(s, a) = r(s, a) + \mathbb{E}_{s' \sim p(\cdot|s,a), a' \sim \pi(\cdot|s')}[\hat{\gamma}(s, a, s')Q_\pi(s', a')]. \tag{4}$$

## 3 OPBRL WITH TRANSITION-DEPENDENT DISCOUNTING

TED starts from the popular three-phase OPBRL framework that first learns a reward model $\hat{r}$ from preference labels, then acquires a labeled dataset $D_{\hat{r}} = \{(s_i, a_i, \hat{r}(s_i, a_i), s_{i+1})\}_{i=1}^N$ with the reward model, and finally performs standard offline RL algorithms on $D_{\hat{r}}$. Section 3.1 shows how TED incorporates the learned reward model's ability to precisely predict low-preference regions with offline RL's pessimism property. TED adds an additional phase before the final phase of applying the offline RL algorithm. It replaces the original discount factor with transition-dependent discount factors according to the predicted data preferences. More specifically, TED assigns lower discount factors to predicted low-preference regions, which decreases these regions' Q values and discourages the agent from visiting these regions. Section 3.2 presents a didactic example that shows how transition-dependent discounting assigns pessimism to the predicted low-quality regions. Finally, Section 3.3 presents a practical implementation of TED.

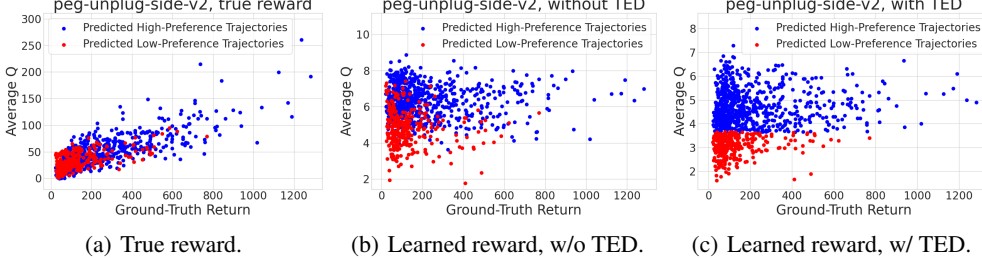

(a) True reward.     (b) Learned reward, w/o TED.     (c) Learned reward, w/ TED.

Figure 2: Visualizations of the Q values learned by an offline RL algorithm IQL, when (a) ground-truth reward is used, (b) learned reward is used w/o TED, and (c) learned reward is used w/ TED. As shown in (a), the predicted low-preference regions (red dots) have low Q values. But in (b), these regions have large Q values due to reward prediction error and Bellman update. In (c), TED decreases these regions' Q values by adjusting discount factors, which makes the agent pessimistic about future returns in the predicted low-preference regions.

## 3.1 TRANSITION-DEPENDENT DISCOUNTING WITH THE REWARD MODEL

As discussed in Section 1 and demonstrated in Figure 1, the reward model learned by OPBRL can predict low-preference regions precisely with very few queries, as the prediction suffers less from generalization errors. Note that popular offline RL algorithms obtain a pessimism property that constrains the agent from staying within dataset support (Kumar et al., 2020; Kostrikov et al., 2021; Cheng et al., 2022), TED extends this pessimism to further discourage the agent from visiting the predicted low-preference regions with transition-dependent discount factors.

Given a dataset $D_{\hat{r}}$ labeled by the reward model $\hat{r}$, TED first finds the trajectories with top $k\%$ returns and forms a new dataset $D_{topk\%}^{\hat{r}}$ that consists of state-action pairs of these top $k\%$ trajectories, where $k \in [0, 100]$ is a hyper-parameter. Then TED computes a transition-dependent discount factor $\gamma_{\hat{r}}$ as follows:

$$\gamma_{\hat{r}}(s, a) = \begin{cases} \gamma, & (s, a) \in D_{topk\%}^{\hat{r}} \\ \sigma * \gamma, & (s, a) \notin D_{topk\%}^{\hat{r}} \end{cases}, \tag{5}$$

where $\sigma \in [0, 1]$ is a hyper-parameter controlling the discount factor of the low-preference regions. $\gamma_{\hat{r}}$ assigns lower discount factors to the predicted low-preference regions, which decreases these regions' Q values. Then, TED replaces the offline algorithm's original discount factor $\gamma$ with $\gamma_{\hat{r}}$, and finally applies the modified offline RL algorithm to the dataset. The pseudo-code for TED is demonstrated in Alg. 1. The modifications TED made to the original OPBRL framework are marked in red.

## 3.2 DIDACTIC EXAMPLE

This subsection proposes a didactic example that demonstrates how the transition-dependent discount constrains the agent from visiting the predicted low-preference regions. Figure 2 visualizes the Q values learned by applying IQL (Kostrikov et al., 2021) to datasets with ground-truth rewards and the learned reward demonstrated in Figure 1, respectively. As shown in Figure 2(a), under ground-truth rewards, the low-preference regions predicted by the learned reward model have low Q values. In contrast, as demonstrated in Figure 2(b), when learned rewards are used and TED is not used, although the predicted low-preference regions have small rewards, they can still obtain large Q values, which may lead the offline agent to visit these regions. This is because offline RL algorithms' Bellman update implicitly stitches trajectories (Sutton & Barto, 2018), and back-propagate erroneous large Q values (the upper-left part of Figure 2(b)) to these predicted low-preference regions. These erroneous Q values are because the learned reward model mistakenly assigns high rewards to ground-truth low-preference regions (the right parts of the subfigures in Figure 1). As shown in Figure 2(c), TED solves this problem by adding transition-dependent discounting to these regions. The agent becomes more pessimistic about future returns in these regions, and the Q values

of the predicted low-preference regions are effectively reduced. The pessimism introduced by TED enables better alignment between Q values and preference predictions, discourages the agent from visiting low-preference regions, and leads to performance improvement over baseline algorithms as demonstrated in Table 1.

### 3.3 A PRACTICAL IMPLEMENTATION

This subsection introduces a practical implementation of TED to current OPBRL algorithms. For the first phase of reward learning, we choose OPRL (Shin et al., 2022) for its simplicity and effectiveness. OPRL learns an ensemble of reward models by iteratively learning the reward model and querying the expert and uses ensemble disagreement to select new queries: it randomly samples a large number of segmentation pairs from the dataset, computes the fraction $p$ of the ensembles that predicts the label $y = 1$, and then computes corresponding prediction variance $p(1 - p)$. Then, OPRL selects the segmentation pair with the highest variance as its query. OPRL's reward model is optimized to minimize the cross-entropy loss defined in Eq. 3.

After learning the reward model, OPRL labels the dataset with the reward model. Then, we perform TED and acquire the transition-dependent discount factor as described in Section 3.1. As for the final phase of offline learning, we replace OPRL's original AWR (Peng et al., 2019) with the more recent IQL algorithm (Kostrikov et al., 2021) for its SOTA performance in offline RL. For offline RL, IQL's loss functions are defined as follows:

$$
\begin{aligned}
L_Q(\theta) &= \mathbb{E}_{(s,a,s') \sim D} \left[ (r(s,a) + \gamma V_\omega(s') - Q_\theta(s,a))^2 \right] \\
L_V(\omega) &= \mathbb{E}_{(s,a) \sim D} \left[ L_2^\tau (Q_{\hat{\theta}}(s,a) - V_\omega(s)) \right] \\
L_\pi(\phi) &= \mathbb{E}_{(s,a) \sim D} \left[ \exp\left( \beta(Q_{\hat{\theta}}(s,a) - V_\omega(s)) \right) \log \pi_\phi(a|s) \right],
\end{aligned}
\tag{6}
$$

where $\theta$, $\hat{\theta}$, $\omega$, and $\phi$ are the parameters of the Q network, the target Q network, the value network, and the policy, respectively, $L_2^\tau(\cdot)$ is the $\tau$ expectile defined as $L_2^\tau(u) = |\tau - \mathbb{1}(u < 0)|u^2$ for any value $u$, and $\beta \in [0, \infty), \tau \in (0, 1)$ are two hyper-parameters. In OPBRL, when we use the reward model $\hat{r}$ to label the dataset $D$ and incorporate TED with IQL, the Q loss becomes:

$$
L_Q^{TED}(\theta) = \mathbb{E}_{(s,a,s') \sim D} \left[ (\hat{r}_\psi(s,a) + \gamma_{\hat{r}}(s,a) V_\omega(s') - Q_\theta(s,a))^2 \right],
\tag{7}
$$

while the value loss $L_V$ and the policy loss $L_\pi$ remain the same.

---

**Algorithm 1** TED: Transition-Dependent Discounting

1: **Require:** An unlabelled offline dataset $D$, an expert $\mathcal{E}$ that can be queried for $N_p$ times, an algorithm $\mathbb{A}_r$ that selects queries and learns a reward model from preference labels, a pessimistic offline RL algorithm $\mathbb{A}_{offline}$, hyper-parameters $k\%$ and $\sigma$
2: Learn a reward model $\hat{r}$ by applying $\mathbb{A}_r$ on $D$ and querying $\mathcal{E}$ {***Phase 1: reward model learning***}
3: Label $D$'s rewards with $\hat{r}$ {***Phase 2: reward labeling***}
4: Compute $\gamma_{\hat{r}}$ according to Eq. 5 and replace the original discount $\gamma$ with $\gamma_{\hat{r}}$ {***Phase 3: Transition-Dependent Discounting***}
5: Apply $\mathbb{A}_{offline}$ to the labelled dataset {***Phase 4: offline policy learning***}

---

## 4 EXPERIMENTS

In this section, we aim to answer the following questions:

1. Can TED outperform SOTA OPBRL algorithms?

2. Is TED sensitive to hyper-parameters? Will it harm performance if the reward is inaccurate? Is TED widely applicable to a range of OPBRL algorithms?

3. Does reward quality matter for D4RL tasks?

We average all the results across six random seeds and present the mean performance and standard variance of the algorithms. Following previous works (Shin et al., 2022; Kang et al., 2023; Lee

et al., 2021; Liang et al., 2021), for all the empirical evaluations in this section, human preferences are simulated by ground-truth rewards, and preference labels are given by comparing segmentations' ground-truth returns.

## 4.1 TED IMPROVES CURRENT OPBRL ALGORITHMS

To acquire a challenging benchmark for evaluating OPBRL algorithms, we create medium-random quality datasets based on the Meta-World MT1 tasks (Yu et al., 2020). The datasets are created by adding random action noises and $\epsilon$-greedy random actions to the scripted policies. Results in Table 2 demonstrate that these tasks are challenging and require high-quality reward labels, as they cannot be solved with simple methods like Top 10% BC or random reward labeling. We also find that OTR, which labels the dataset with optimal transport (OT) distances to expert demonstrations (Luo et al., 2022), also fails to solve these tasks, as Meta-World tasks are complex and OT distances cannot predict rewards precisely.

Table 3 demonstrates algorithms' average performance on the 50 Meta-World tasks, and Table 4 shows detailed algorithm performance on 10 example tasks. We compare TED against IQL with true rewards (True Reward), as well as three baseline algorithms OPRL-I, PT, and OPPO. OPRL-I replaces OPRL's original offline RL algorithm AWR with the more advanced IQL algorithm and learns rewards by selecting queries with large ensemble disagreement (Shin et al., 2022). In contrast, PT learns a weighted sum of non-Markovian rewards with Transformers (Vaswani et al., 2017). Both baselines use IQL as the offline RL algorithm. OPPO (Kang et al., 2023) learns a representation of preferences and uses this representation as context for the policy. Results show that TED can significantly improve baselines' performance when the number of preference queries is small (10). In contrast, baseline algorithms OPRL-I and PT fail to achieve good performance as the learned reward is inaccurate. OPPO's performance is lower than PT, as learning a good representation is query-inefficient. Overall, PT+TED achieves true-reward-level performance on 13 out of the 50 tasks and has notable improvement over PT on 26 tasks. The remaining tasks that TED does not have notable improvement in are either too hard to learn even with true rewards or are so easy that they can be solved with baseline algorithms. Detailed performance on all 50 tasks is deferred to Appendix B.1.

Table 2: Comparison between IQL with different reward labeling schemes and the Top 10% BC baseline. The MT1 tasks cannot be solved with simple reward labeling techniques, imitating the top 10% trajectories, or labeling reward with optimal transport distance to expert demonstrations. Scores are normalized by the maximum ground-truth return in each task's corresponding dataset.

| True Reward | Random Reward | Top 10% BC | OTR |
|---|---|---|---|
| **0.65±0.07** | 0.22±0.04 | 0.20±0.01 | 0.06±0.01 |

Table 3: Algorithms' normalized scores averaged over 50 Meta-World ML1 tasks.

| True Reward | OPRL-I | OPRL-I+TED | PT | PT+TED | OPPO |
|---|---|---|---|---|---|
| **0.65±0.07** | 0.33±0.02 | 0.39±0.03 | 0.27±0.03 | 0.40±0.04 | 0.19±0.02 |

## 4.2 ABLATION STUDIES

**Hyper-parameter Sensitivity** TED introduces two additional hyper-parameter, the threshold $k\%$ and the discount factor weight $\sigma$. For all the experiments in Section 4.1, we use the same set of hyper-parameters ($k = 70, \sigma = 0.7$), which indicates that TED is generally applicable to a wide range of tasks without particular fine-tuning. We choose OPRL-I as the baseline algorithm and further compare OPRL-I+TED with various hyper-parameters. Results show that TED is robust to the choice of hyper-parameters and achieves stable performance. Detailed algorithm performance is deferred to Appendix B.2.1.

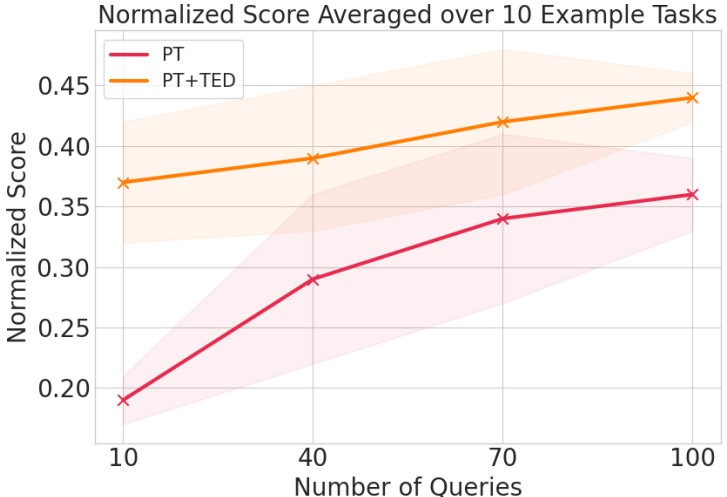

Figure 3: Performance of PT and PT+TED with different reward model qualities. TED substantially outperforms the original OPBRL baseline with different numbers of queries, and does not harm performance if ground-truth rewards are used. Note that PT+TED with 10 queries achieves comparable performance to PT with 100 queries.

Table 4: Performance of TED and various baselines on 10 example MT1 tasks. All OPBRL methods use 10 queries. Underlined data indicates that TED has significant improvement over the original OPBRL baseline.

| Example Task | True Reward | OPRL-I | OPRL-I+TED | PT | PT+TED |
|---|---|---|---|---|---|
| box-close-v2 | **0.94±0.01** | 0.14±0.00 | 0.41±0.03 | 0.15±0.00 | 0.34±0.12 |
| drawer-open-v2 | **0.76±0.01** | 0.59±0.02 | 0.56±0.02 | 0.13±0.00 | 0.64±0.04 |
| hammer-v2 | **1.22±0.06** | 0.30±0.01 | 0.43±0.03 | 0.23±0.00 | 0.45±0.04 |
| handle-pull-side-v2 | 0.27±0.14 | 0.30±0.06 | **0.47±0.04** | 0.03±0.00 | 0.10±0.03 |
| pick-place-v2 | **0.53±0.16** | 0.01±0.00 | 0.14±0.01 | 0.01±0.00 | 0.44±0.08 |
| plate-slide-back-v2 | 0.23±0.02 | 0.22±0.02 | 0.25±0.04 | 0.20±0.03 | 0.19±0.00 |
| push-v2 | **0.78±0.16** | 0.03±0.01 | 0.23±0.05 | 0.03±0.02 | 0.18±0.14 |
| reach-wall-v2 | **0.91±0.01** | 0.81±0.01 | 0.79±0.01 | 0.85±0.02 | 0.85±0.01 |
| sweep-v2 | **0.88±0.00** | 0.08±0.02 | 0.49±0.12 | 0.18±0.16 | 0.43±0.04 |
| sweep-into-v2 | **0.76±0.00** | 0.25±0.04 | 0.67±0.01 | 0.10±0.00 | 0.11±0.01 |

**Reward Model Quality**   Figure 3 demonstrates TED's ability to improve over baseline algorithms when reward quality changes, tested over the 10 example tasks listed in Table 4. We use PT as the baseline and test algorithms with reward models trained with different numbers of queries as well as the true rewards. Results show that PT+TED consistently outperforms TED with different reward model qualities, and does not harm performance if ground-truth rewards are used: PT with ground-truth reward achieves an average performance of $0.72 \pm 0.06$, while PT+TED with ground-truth reward achieves $0.71 \pm 0.05$. We also observe that PT+TED with 10 queries can perform comparable to the original PT with 100 queries, demonstrating TED's ability to learn policies with high query efficiency. Detailed algorithm performance on these 10 tasks is deferred to Appendix B.2.2.

**Offline Algorithm**   To demonstrate that TED is generally applicable to a series of pessimistic offline RL algorithms, we test TED with a variant of OPRL that uses another popular offline RL algorithm, ATAC (Cheng et al., 2022) as the offline algorithm. This OPRL variant is named OPRL-ATAC. Results demonstrate that OPRL-ATAC+TED also improves over OPRL-ATAC, which supports our claim. Detailed algorithm performance is deferred to Appendix B.2.3.

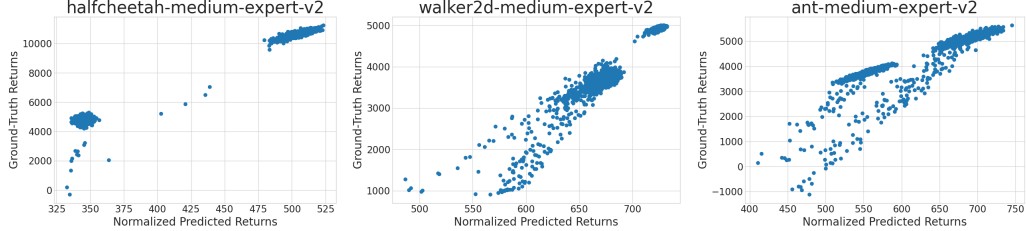

Figure 4: Visualization of the correspondence between returns predicted by OPRL-I and ground-truth returns on 3 D4RL MuJoCo tasks. OPRL-I can learn a precise prediction of rewards with merely 10 queries.

**Truncating the Dataset**  We test another way to utilize the reward model's precise prediction on the low-preference regions, which simply discards low-preference data and only uses $D_{topk\%}$ for offline RL learning. This variant, named "PT+DatasetTruncating", even underperforms the original PT algorithm, only achieving an average performance of $0.18 \pm 0.02$ while the original PT achieves $0.27 \pm 0.03$. This performance difference is because DatasetTruncating does not explicitly discourage the agent from visiting low-preference regions and instead loses information about the quality of these regions by discarding data. Detailed results on all the 50 tasks are deferred to Appendix B.2.4.

### 4.3 D4RL IS ROBUST TO REWARD QUALITY

Previous OPBRL works (Shin et al., 2022; Kim et al., 2022; Kang et al., 2023) use D4RL (Fu et al., 2020) as their evaluation environment. However, this subsection demonstrates that most D4RL tasks can be solved with a simple baseline and merely 10 queries, indicating that these tasks cannot effectively reflect OPBRL algorithm efficiency. The main reasons are: 1) D4RL MuJoCo tasks obtain simple reward functions that are easy to fit with few preference labels, and 2) most D4RL datasets are of high quality, which enables pessimistic offline RL algorithms to achieve high performance with inaccurate rewards (Li et al., 2023).

We use OPRL-I as the baseline algorithm and test it on 26 D4RL tasks that consist of 5 hopper tasks, 5 halfcheetah tasks, 5 walker2d tasks, 5 ant tasks, and 6 antmaze tasks. Experiment results in Table 6 demonstrate that this baseline only needs 10 queries to achieve true-reward-level performance on various D4RL tasks. The first reason for this surprising result is that D4RL MuJoCo tasks obtain simple reward functions that are highly correlated to a single state dimension and are easy to predict. Table 5 and Figure 4 demonstrate that for these tasks, the reward is almost a linear function of a certain state dimension and, therefore, can be estimated well with 10 queries. The second reason is that most D4RL datasets are high-quality and insensitive to reward quality. Table 6 demonstrates that some tasks can be solved even with random rewards. This result corresponds to Li et al. (2023), which claims that pessimistic offline RL algorithms enjoy a "survival instinct" that makes them insensitive to rewards if the dataset is of high quality, as these algorithms constrain the agent to stay within the dataset support. Based on these observations above, we suggest that most D4RL tasks are not sensitive to reward quality and may not serve as a proper benchmark for evaluating OPBRL. Full results on all 26 tasks are deferred to Appendix B.3.

Table 5: Correlation between true reward and state, as well as the correlation between predicted returns and true returns on 3 D4RL MuJoCo tasks.

| Task | Correlation between true reward and one certain dimension of the state | Correlation between predicted returns and true returns |
|---|---|---|
| halfcheetah-medium-expert-v2 | 0.990 | 0.995 |
| walker2d-medium-expert-v2 | 0.998 | 0.971 |
| ant-medium-expert-v2 | 0.943 | 0.870 |

Table 6: Comparison between IQL with true reward, IQL with random reward, and IQL with reward model learned with OPRL and 10 queries. OPRL-I achieves true-reward level performance with merely 10 queries.

| D4RL Task | True Reward | Random Reward | 10 Queries |
|---|---|---|---|
| hopper-medium-expert-v2 | **0.98±0.09** | 0.71±0.03 | **1.09±0.01** |
| hopper-medium-v2 | **0.54±0.02** | **0.53±0.00** | **0.55±0.01** |
| halfcheetah-medium-expert-v2 | **0.85±0.03** | 0.64±0.02 | 0.70±0.05 |
| halfcheetah-medium-v2 | **0.43±0.00** | **0.42±0.00** | **0.42±0.00** |
| ant-medium-expert-v2 | **1.27±0.01** | 1.21±0.02 | 1.16±0.02 |
| ant-medium-v2 | **0.90±0.02** | **0.90±0.02** | **0.90±0.01** |
| walker2d-medium-expert-v2 | **1.09±0.00** | **1.09±0.00** | **1.09±0.00** |
| walker2d-medium-v2 | **0.80±0.01** | 0.78±0.02 | 0.75±0.02 |
| antmaze-medium-diverse-v2 | 0.68±0.08 | 0.27±0.14 | **0.82±0.03** |
| antmaze-large-diverse-v2 | **0.43±0.03** | 0.05±0.02 | **0.39±0.06** |
| antmaze-umaze-diverse-v2 | **0.68±0.02** | 0.27±0.09 | 0.32±0.20 |
| Average over 26 tasks | **0.69±0.03** | 0.57±0.03 | **0.69±0.03** |

## 5 RELATED WORK

**OPBRL.** Current OPBRL methods mainly follow two learning schemes. OPRL (Shin et al., 2022) and PT (Kim et al., 2022) adopt a pipeline framework of first learning a reward model from preferences, then labeling the dataset with the learned reward model, and finally applying standard offline RL algorithms. OPRL applies uncertainty quantifications commonly used in active learning literature for query selection, while PT learns a weighted sum of non-Markovian rewards with Transformers. OPPO (Kang et al., 2023) models preference prediction as a representation learning problem and conditions the policy with the learned representations. DPO (Rafailov et al., 2023) models task generation as a one-step MDP and performs end-to-end policy optimization using a learning algorithm similar to AWR (Peng et al., 2019).

**Transition-dependent discounting.** In RL, transition-dependent or state-dependent discounting, although a relatively nascent technique, has been incorporated into numerous prevalent algorithms, such as the Emphatic Temporal Difference learning algorithm (ETD)(Mahmood et al., 2015) for off-policy temporal prediction, for generalizing task formalism(White, 2017; Silver et al., 2017a; Pitis, 2019), the ExQ-learning algorithm(Yoshida et al., 2013) for fast learning, and dynamic programming in economics (Wei & Guo, 2011; Stachurski & Zhang, 2021). Neurobiology studies also suggest animals are likely to regulate the reward discounting depending on their situations(Miyazaki et al., 2012). Contrasted with conventional fixed-discounted approaches, transition-dependent discounting, by emphasizing certain transitions more probable to yield higher outcomes, facilitates enhanced and generalizable representations of real-world dynamics, superior adaptability to complex environments, accelerated learning speed, and improved task specification's flexibility and generalization.

**Reward labeling in offline RL.** Several recent works focus on the problem of labeling rewards of offline datasets. OTR (Luo et al., 2022) assumes that several expert demonstrations are available and uses optimal transport (Villani et al., 2009) to label rewards. Li et al. (2023) claims that pessimistic offline RL algorithms are not sensitive to reward quality if the datasets are high quality and can achieve comparable performance with trivial or even wrong rewards on high-quality datasets.

## 6 CONCLUSION

This paper proposes and discusses the problem of achieving query-efficient OPBRL. We propose a novel and conceptually simple OPBRL method, TED, which is applicable to a wide range of OPBRL algorithms and achieves query-efficient learning. We also propose a challenging OPBRL benchmark based on Meta-World MT1 and demonstrate TED's ability to improve current OPBRL algorithms notably. An interesting future direction is to incorporate TED with real preferences from humans, which can be noisy and occasionally erroneous.

## REPRODUCIBILITY STATEMENT

Our detailed algorithm implementation is demonstrated in Section 3. Hyper-parameter settings as well as dataset generation procedure are demonstrated in Appendix A. The code for reproducing TED is included in our supplementary materials.

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

## A    HYPER-PARAMETER SETTINGS

We use the same hyper-parameter set for all TED experiments: $k = 70$, and $\sigma = 0.7$. Each Meta-World dataset consists of 50 trajectories collected by the corresponding scripted policy added with Gaussian noise with mean 0 and std 0.8, and 950 trajectories collected with a policy that adds $\epsilon$-greedy random actions to the former noisy policy, where $\epsilon = 0.8$. Table 7 shows the average and maximum return of the 50 datasets. For OPRL-I, we use 7 ensembles. Each ensemble is initially trained with 1 randomly selected query and then performs 3 rounds of active querying and training, and in each round, 1 query is acquired, making a total of 10 queries. For PT, we follow its original hyper-parameter settings, and change the number of queries to 10. For IQL, we use the default hyper-parameter settings: 3e-4 for all learning rates, $\tau = 0.7$, and set the temperature $\beta$ to 3.0. For ATAC, we follow its original hyper-parameter settings, and set $\beta$ to 4.0.

## B    ADDITIONAL EXPERIMENT RESULTS

### B.1    DETAILED ALGORITHM PERFORMANCE ON META-WORLD

Table 8 and Table 9 demonstrate algorithms' performance on all the 50 Meta-World MT1 tasks. TED significantly improves over baseline algorithms OPRL-I and PT.

### B.2    DETAILED RESULTS FOR ABLATION STUDIES

This subsection demonstrates detailed results for ablations studies in Section 4.2.

#### B.2.1    DETAILED RESULTS FOR HYPER-PARAMETER ABLATIONS

Results in Table 10 demonstrate hyper-parameter ablation results on the 10 example tasks and show that TED is generally robust to the choice of hyper-parameters.

#### B.2.2    DETAILED RESULTS FOR REWARD QUALITY ABLATION

Table 11 and Table 12 show the improvement of PT+TED over PT on the 10 example tasks. Results show that PT+TED consistently outperforms PT with different reward qualities and does not degrade performance if ground-truth rewards are used. Results for the "10 queries" row are demonstrated in Table 4.

#### B.2.3    DETAILED RESULTS FOR TED WITH ATAC

Results in Table 13 demonstrate the improvement of OPRL-ATAC+TED over OPRL with ATAC, which proves that TED is generally applicable to a series of pessimistic offline RL algorithms.

#### B.2.4    DETAILED RESULTS FOR PT+DATASETTRUNCATING

Table 14 shows PT+DatasetTruncating's results on all 50 Meta-World tasks. This variant fails to perform well as it discards information about low-preference regions.

### B.3    DETAILED RESULTS FOR D4RL

Table 15 demonstrates algorithms' performance on 26 D4RL tasks. These tasks can be solved with merely 10 queries.

Table 7: Average and maximum returns of the 50 datasets we use for empirical evaluation.

| Task | Average Dataset Return | Maximum Dataset Return |
|---|---|---|
| assembly-v2 | 370.52 | 3885.59 |
| basketball-v2 | 170.15 | 4174.84 |
| bin-picking-v2 | 48.32 | 2536.60 |
| box-close-v2 | 515.10 | 3927.44 |
| button-press-topdown-v2 | 1551.68 | 3867.07 |
| button-press-v2 | 218.22 | 543.88 |
| coffee-button-v2 | 352.38 | 1840.48 |
| coffee-pull-v2 | 177.47 | 4055.55 |
| coffee-push-v2 | 174.72 | 3485.67 |
| disassemble-v2 | 254.29 | 4018.66 |
| door-close-v2 | 2951.36 | 4572.39 |
| door-lock-v2 | 734.16 | 3550.99 |
| door-open-v2 | 1585.94 | 4404.95 |
| door-unlock-v2 | 2766.30 | 4512.69 |
| drawer-close-v2 | 2317.97 | 4893.05 |
| drawer-open-v2 | 1329.73 | 4095.00 |
| faucet-close-v2 | 3174.24 | 4645.53 |
| faucet-open-v2 | 3168.57 | 4667.57 |
| hammer-v2 | 525.90 | 1851.67 |
| hand-insert-v2 | 216.37 | 4585.00 |
| handle-press-side-v2 | 752.99 | 4447.10 |
| handle-press-v2 | 999.01 | 4838.08 |
| handle-pull-side-v2 | 419.45 | 2797.47 |
| handle-pull-v2 | 763.41 | 4158.05 |
| lever-pull-v2 | 560.14 | 2052.01 |
| peg-insert-side-v2 | 219.53 | 4308.08 |
| peg-unplug-side-v2 | 197.61 | 1282.24 |
| pick-out-of-hole-v2 | 34.81 | 2136.05 |
| pick-place-v2 | 74.64 | 3879.25 |
| pick-place-wall-v2 | 22.44 | 3612.44 |
| plate-slide-back-side-v2 | 663.53 | 4399.59 |
| plate-slide-back-v2 | 453.59 | 2947.07 |
| plate-slide-side-v2 | 1032.45 | 4543.43 |
| plate-slide-v2 | 1023.59 | 4605.91 |
| push-back-v2 | 42.50 | 1940.71 |
| push-v2 | 85.99 | 3880.29 |
| push-wall-v2 | 74.59 | 3465.33 |
| reach-v2 | 3457.47 | 4807.45 |
| reach-wall-v2 | 3428.71 | 4762.16 |
| soccer-v2 | 957.92 | 4566.24 |
| stick-push-v2 | 64.08 | 2364.95 |
| sweep-v2 | 477.99 | 4397.31 |
| sweep-into-v2 | 1175.53 | 4663.26 |
| window-close-v2 | 1607.88 | 3808.67 |
| window-open-v2 | 839.56 | 2867.00 |
| button-press-topdown-wall-v2 | 225.09 | 782.20 |
| dial-turn-v2 | 676.19 | 3335.82 |
| button-press-wall-v2 | 573.05 | 3005.26 |
| shelf-place-v2 | 111.67 | 3803.59 |
| stick-pull-v2 | 71.21 | 3944.00 |

Table 8: Performance of various baselines on 50 Meta-World MT1 tasks. All OPBRL methods use 10 queries. Simple techniques like Random Reward or Top 10% BC fail to solve these tasks. OTR also fails as Meta-World tasks are complex.

| Task | True Reward | Random Reward | Top 10% BC | OTR | OPPO |
|---|---|---|---|---|---|
| assembly-v2 | **0.25±0.13** | 0.06±0.00 | 0.09±0.00 | 0.05±0.00 | 0.07±0.01 |
| basketball-v2 | **0.90±0.00** | 0.73±0.04 | 0.11±0.01 | 0.00±0.00 | 0.00±0.00 |
| bin-picking-v2 | **1.12±0.01** | 0.02±0.01 | 0.06±0.01 | 0.00±0.00 | 0.01±0.00 |
| box-close-v2 | **0.94±0.01** | 0.11±0.00 | 0.12±0.00 | 0.07±0.00 | 0.09±0.01 |
| button-press-topdown-v2 | **0.75±0.00** | 0.58±0.01 | 0.33±0.01 | 0.36±0.24 | 0.09±0.03 |
| button-press-v2 | **0.92±0.00** | 0.81±0.01 | 0.40±0.01 | 0.15±0.01 | 0.27±0.05 |
| coffee-button-v2 | **0.30±0.03** | 0.10±0.01 | 0.17±0.01 | 0.06±0.00 | 0.19±0.03 |
| coffee-pull-v2 | **0.54±0.04** | 0.07±0.03 | 0.04±0.00 | 0.01±0.00 | 0.01±0.00 |
| coffee-push-v2 | **0.81±0.04** | 0.01±0.00 | 0.03±0.00 | 0.00±0.00 | 0.03±0.01 |
| disassemble-v2 | **0.53±0.01** | 0.06±0.00 | 0.06±0.00 | 0.05±0.00 | 0.05±0.00 |
| door-close-v2 | **0.80±0.00** | 0.34±0.02 | 0.22±0.19 | 0.01±0.00 | 0.38±0.13 |
| door-lock-v2 | 0.78±0.07 | 0.75±0.02 | 0.33±0.02 | 0.19±0.03 | 0.15±0.01 |
| door-open-v2 | **0.58±0.05** | 0.28±0.08 | 0.30±0.04 | 0.09±0.01 | 0.05±0.00 |
| door-unlock-v2 | **0.84±0.02** | 0.76±0.01 | 0.51±0.01 | 0.19±0.04 | 0.11±0.01 |
| drawer-close-v2 | **0.53±0.00** | 0.23±0.10 | 0.38±0.04 | 0.01±0.01 | 0.26±0.19 |
| drawer-open-v2 | **0.76±0.01** | 0.19±0.01 | 0.32±0.02 | 0.09±0.00 | 0.11±0.00 |
| faucet-close-v2 | **0.77±0.04** | 0.58±0.08 | 0.59±0.01 | 0.19±0.07 | 0.30±0.03 |
| faucet-open-v2 | **0.78±0.01** | 0.52±0.06 | 0.59±0.01 | 0.35±0.05 | 0.28±0.03 |
| hammer-v2 | **1.22±0.06** | 0.25±0.02 | 0.29±0.01 | 0.26±0.00 | 0.26±0.00 |
| hand-insert-v2 | **0.80±0.01** | 0.04±0.04 | 0.04±0.01 | 0.00±0.00 | 0.00±0.00 |
| handle-press-side-v2 | **0.57±0.30** | 0.20±0.07 | 0.18±0.02 | 0.02±0.01 | 0.16±0.03 |
| handle-press-v2 | **0.97±0.00** | 0.29±0.01 | 0.26±0.03 | 0.23±0.03 | 0.18±0.04 |
| handle-pull-side-v2 | **0.27±0.14** | 0.18±0.10 | 0.14±0.02 | 0.00±0.00 | 0.02±0.01 |
| handle-pull-v2 | **0.29±0.18** | 0.04±0.01 | 0.15±0.01 | 0.00±0.00 | 0.04±0.01 |
| lever-pull-v2 | 0.41±0.04 | 0.43±0.05 | 0.26±0.00 | 0.11±0.02 | 0.13±0.02 |
| peg-insert-side-v2 | **0.79±0.01** | 0.00±0.00 | 0.10±0.01 | 0.00±0.00 | 0.00±0.00 |
| peg-unplug-side-v2 | **0.37±0.10** | 0.24±0.05 | 0.10±0.01 | 0.01±0.00 | 0.05±0.01 |
| pick-out-of-hole-v2 | **1.23±0.15** | 0.00±0.00 | 0.01±0.00 | 0.00±0.00 | 0.00±0.00 |
| pick-place-v2 | **0.53±0.16** | 0.00±0.00 | 0.02±0.00 | 0.00±0.00 | 0.01±0.00 |
| pick-place-wall-v2 | **0.04±0.01** | 0.00±0.00 | 0.00±0.00 | 0.00±0.00 | 0.00±0.00 |
| plate-slide-back-side-v2 | **0.36±0.27** | 0.04±0.01 | 0.16±0.01 | 0.01±0.00 | 0.05±0.01 |
| plate-slide-back-v2 | **0.23±0.02** | 0.15±0.01 | 0.16±0.01 | 0.01±0.00 | 0.07±0.02 |
| plate-slide-side-v2 | 0.30±0.03 | 0.43±0.10 | 0.20±0.00 | 0.00±0.00 | 0.27±0.03 |
| plate-slide-v2 | **0.86±0.01** | 0.09±0.01 | 0.24±0.03 | 0.02±0.00 | 0.08±0.01 |
| push-back-v2 | **1.15±0.05** | 0.01±0.00 | 0.04±0.01 | 0.00±0.00 | 0.00±0.00 |
| push-v2 | **0.78±0.16** | 0.20±0.21 | 0.03±0.00 | 0.00±0.00 | 0.01±0.00 |
| push-wall-v2 | **0.97±0.20** | 0.01±0.00 | 0.02±0.00 | 0.01±0.00 | 0.01±0.00 |
| reach-v2 | **0.95±0.00** | 0.52±0.04 | 0.51±0.03 | 0.12±0.07 | 0.58±0.06 |
| reach-wall-v2 | **0.91±0.01** | 0.69±0.04 | 0.55±0.01 | 0.04±0.00 | 0.29±0.07 |
| soccer-v2 | **0.42±0.17** | 0.21±0.07 | 0.12±0.00 | 0.05±0.01 | 0.05±0.01 |
| stick-push-v2 | **0.33±0.22** | 0.01±0.00 | 0.02±0.00 | 0.00±0.00 | 0.01±0.00 |
| sweep-v2 | **0.88±0.00** | 0.11±0.07 | 0.18±0.01 | 0.00±0.00 | 0.03±0.00 |
| sweep-into-v2 | **0.76±0.00** | 0.27±0.13 | 0.19±0.02 | 0.00±0.00 | 0.05±0.01 |
| window-close-v2 | **0.47±0.07** | 0.12±0.00 | 0.32±0.02 | 0.04±0.01 | 0.09±0.01 |
| window-open-v2 | **0.43±0.00** | 0.10±0.00 | 0.25±0.01 | 0.07±0.01 | 0.08±0.01 |
| button-press-topdown-wall-v2 | **0.18±0.09** | 0.00±0.00 | 0.25±0.02 | 0.05±0.02 | 0.06±0.04 |
| dial-turn-v2 | **0.70±0.04** | 0.09±0.07 | 0.18±0.01 | 0.00±0.00 | 0.09±0.01 |
| button-press-wall-v2 | **0.75±0.02** | 0.21±0.21 | 0.21±0.01 | 0.02±0.00 | 0.05±0.01 |
| shelf-place-v2 | **0.28±0.29** | 0.00±0.00 | 0.02±0.01 | 0.00±0.00 | 0.00±0.00 |
| stick-pull-v2 | **0.18±0.01** | 0.01±0.00 | 0.02±0.01 | 0.00±0.00 | 0.00±0.00 |
| Average | **0.65±0.07** | 0.22±0.04 | 0.20±0.01 | 0.06±0.01 | 0.19±0.02 |

Table 9: Performance of TED and various baselines on 50 Meta-World MT1 tasks. All OPBRL methods use 10 queries. The underlined data represents that TED has significant improvement over the original algorithm.

| Task | True Reward | OPRL-I | OPRL-I+TED | PT | PT+TED |
|---|---|---|---|---|---|
| assembly-v2 | **0.25±0.13** | 0.15±0.01 | 0.15±0.01 | 0.10±0.00 | **0.30±0.04** |
| basketball-v2 | **0.90±0.00** | 0.07±0.05 | **0.80±0.05** | 0.00±0.00 | **0.84±0.01** |
| bin-picking-v2 | **1.12±0.01** | 0.84±0.13 | 0.73±0.14 | 0.01±0.00 | 0.87±0.03 |
| box-close-v2 | **0.94±0.01** | 0.14±0.00 | 0.41±0.03 | 0.15±0.00 | 0.34±0.12 |
| button-press-topdown-v2 | **0.75±0.00** | 0.63±0.01 | 0.61±0.01 | 0.66±0.02 | **0.71±0.01** |
| button-press-v2 | **0.92±0.00** | 0.83±0.01 | 0.85±0.00 | 0.68±0.15 | 0.79±0.00 |
| coffee-button-v2 | **0.30±0.03** | 0.24±0.00 | 0.25±0.00 | 0.25±0.00 | 0.24±0.01 |
| coffee-pull-v2 | **0.54±0.04** | 0.01±0.00 | 0.07±0.02 | 0.01±0.00 | 0.01±0.00 |
| coffee-push-v2 | **0.81±0.04** | 0.02±0.00 | 0.20±0.03 | 0.02±0.00 | 0.11±0.10 |
| disassemble-v2 | **0.53±0.01** | 0.08±0.01 | 0.14±0.03 | 0.08±0.01 | 0.09±0.00 |
| door-close-v2 | **0.80±0.00** | 0.68±0.01 | 0.52±0.01 | 0.68±0.02 | 0.65±0.07 |
| door-lock-v2 | **0.78±0.07** | **0.79±0.01** | 0.72±0.02 | 0.74±0.02 | **0.78±0.01** |
| door-open-v2 | **0.58±0.05** | **0.58±0.07** | 0.42±0.16 | 0.24±0.03 | 0.40±0.11 |
| door-unlock-v2 | **0.84±0.02** | 0.79±0.01 | 0.78±0.03 | **0.86±0.01** | 0.80±0.01 |
| drawer-close-v2 | **0.53±0.00** | 0.22±0.04 | 0.31±0.02 | 0.52±0.01 | 0.50±0.01 |
| drawer-open-v2 | **0.76±0.01** | 0.59±0.02 | 0.56±0.02 | 0.13±0.00 | 0.64±0.04 |
| faucet-close-v2 | **0.77±0.04** | **0.78±0.00** | 0.76±0.00 | 0.79±0.03 | **0.80±0.01** |
| faucet-open-v2 | **0.78±0.01** | **0.78±0.01** | 0.74±0.01 | 0.82±0.03 | 0.75±0.02 |
| hammer-v2 | **1.22±0.06** | 0.30±0.01 | 0.43±0.03 | 0.23±0.00 | 0.45±0.04 |
| hand-insert-v2 | **0.80±0.01** | 0.07±0.00 | 0.21±0.03 | 0.01±0.00 | 0.41±0.07 |
| handle-press-side-v2 | **0.57±0.30** | 0.23±0.00 | 0.25±0.01 | 0.24±0.00 | 0.25±0.01 |
| handle-press-v2 | **0.97±0.00** | 0.30±0.02 | 0.29±0.01 | 0.30±0.00 | 0.30±0.00 |
| handle-pull-side-v2 | 0.27±0.14 | 0.30±0.06 | **0.47±0.04** | 0.03±0.00 | 0.10±0.03 |
| handle-pull-v2 | **0.29±0.18** | 0.04±0.00 | 0.05±0.00 | 0.03±0.00 | 0.03±0.00 |
| lever-pull-v2 | 0.41±0.04 | **0.61±0.02** | **0.62±0.01** | 0.28±0.01 | 0.48±0.03 |
| peg-insert-side-v2 | **0.79±0.01** | 0.07±0.01 | 0.14±0.03 | 0.08±0.03 | 0.12±0.03 |
| peg-unplug-side-v2 | **0.37±0.10** | 0.30±0.06 | **0.43±0.04** | 0.25±0.05 | **0.39±0.09** |
| pick-out-of-hole-v2 | **1.23±0.15** | 0.00±0.00 | 0.01±0.00 | 0.35±0.16 | 0.14±0.10 |
| pick-place-v2 | **0.53±0.16** | 0.01±0.00 | 0.14±0.01 | 0.01±0.00 | 0.44±0.08 |
| pick-place-wall-v2 | **0.04±0.01** | 0.00±0.00 | **0.04±0.01** | 0.00±0.00 | **0.04±0.03** |
| plate-slide-back-side-v2 | **0.36±0.27** | 0.16±0.01 | 0.17±0.02 | 0.15±0.00 | 0.15±0.00 |
| plate-slide-back-v2 | 0.23±0.02 | 0.22±0.02 | 0.25±0.04 | 0.20±0.03 | 0.19±0.00 |
| plate-slide-side-v2 | 0.30±0.03 | 0.24±0.01 | **0.50±0.06** | 0.24±0.01 | 0.24±0.00 |
| plate-slide-v2 | **0.86±0.01** | 0.78±0.01 | 0.49±0.08 | 0.12±0.00 | 0.28±0.11 |
| push-back-v2 | **1.15±0.05** | 0.02±0.00 | 0.23±0.05 | 0.01±0.00 | 0.31±0.14 |
| push-v2 | **0.78±0.16** | 0.03±0.01 | 0.23±0.05 | 0.03±0.02 | 0.18±0.14 |
| push-wall-v2 | **0.97±0.20** | 0.12±0.00 | 0.25±0.06 | 0.35±0.37 | **1.04±0.02** |
| reach-v2 | **0.95±0.00** | 0.77±0.02 | 0.80±0.01 | 0.81±0.05 | **0.93±0.01** |
| reach-wall-v2 | **0.91±0.01** | 0.81±0.01 | 0.79±0.01 | 0.85±0.02 | 0.85±0.01 |
| soccer-v2 | **0.42±0.17** | 0.32±0.02 | 0.34±0.00 | **0.39±0.14** | 0.27±0.05 |
| stick-push-v2 | **0.33±0.22** | 0.02±0.00 | 0.04±0.01 | 0.01±0.00 | 0.05±0.05 |
| sweep-v2 | **0.88±0.00** | 0.08±0.02 | 0.49±0.12 | 0.18±0.16 | 0.43±0.04 |
| sweep-into-v2 | **0.76±0.00** | 0.25±0.04 | 0.67±0.01 | 0.10±0.00 | 0.11±0.01 |
| window-close-v2 | 0.47±0.07 | 0.48±0.01 | 0.45±0.01 | **0.52±0.04** | **0.54±0.00** |
| window-open-v2 | **0.43±0.00** | 0.30±0.01 | 0.33±0.01 | 0.19±0.05 | 0.32±0.02 |
| button-press-topdown-wall-v2 | **0.18±0.09** | 0.06±0.01 | 0.05±0.00 | 0.05±0.10 | 0.00±0.00 |
| dial-turn-v2 | **0.70±0.04** | 0.43±0.02 | 0.42±0.01 | 0.15±0.02 | 0.20±0.01 |
| button-press-wall-v2 | **0.75±0.02** | 0.60±0.01 | 0.60±0.01 | 0.68±0.07 | **0.71±0.04** |
| shelf-place-v2 | **0.28±0.26** | 0.09±0.01 | 0.10±0.00 | 0.00±0.00 | 0.11±0.03 |
| stick-pull-v2 | **0.18±0.01** | **0.22±0.03** | 0.18±0.06 | 0.03±0.03 | **0.18±0.01** |
| Average | **0.65±0.07** | 0.33±0.02 | 0.39±0.03 | 0.27±0.03 | 0.40±0.04 |

Table 10: Ablation study on OPRL-I+TED's hyper-parameters $k$ and $\sigma$. Scores are averaged over the 10 example tasks in Table 4.

| Task | True Reward | OPRL-I | $k = 70$ $\sigma = 0.7$ | $k = 50$ $\sigma = 0.7$ | $k = 70$ $\sigma = 0.5$ | $k = 50$ $\sigma = 0.5$ |
|---|---|---|---|---|---|---|
| box-close-v2 | **0.94±0.01** | 0.14±0.00 | 0.41±0.03 | 0.35±0.04 | 0.20±0.00 | 0.38±0.05 |
| drawer-open-v2 | **0.76±0.01** | 0.59±0.02 | 0.56±0.02 | 0.50±0.02 | 0.42±0.03 | 0.53±0.03 |
| hammer-v2 | **1.22±0.06** | 0.30±0.01 | 0.43±0.03 | 0.53±0.04 | 0.47±0.03 | 0.60±0.03 |
| handle-pull-side-v2 | 0.27±0.14 | 0.30±0.06 | 0.47±0.04 | 0.44±0.02 | 0.47±0.03 | **0.55±0.07** |
| sweep-v2 | **0.88±0.00** | 0.08±0.02 | 0.49±0.12 | 0.50±0.03 | 0.43±0.05 | 0.43±0.06 |
| pick-place-v2 | **0.53±0.16** | 0.01±0.00 | 0.14±0.01 | 0.12±0.01 | 0.11±0.03 | 0.11±0.02 |
| plate-slide-back-v2 | 0.23±0.02 | 0.22±0.02 | 0.25±0.04 | 0.26±0.04 | 0.21±0.02 | 0.20±0.01 |
| push-v2 | **0.78±0.16** | 0.03±0.01 | 0.23±0.05 | 0.14±0.04 | 0.12±0.01 | 0.14±0.04 |
| reach-wall-v2 | **0.91±0.01** | 0.81±0.01 | 0.79±0.01 | 0.78±0.01 | 0.78±0.01 | 0.79±0.01 |
| sweep-into-v2 | **0.76±0.00** | 0.25±0.04 | 0.67±0.01 | 0.60±0.03 | 0.50±0.03 | 0.58±0.01 |
| Average | **0.72±0.06** | 0.27±0.02 | 0.44±0.04 | 0.42±0.03 | 0.37±0.03 | 0.43±0.03 |

Table 11: Ablation study on reward model quality (50 and 100 queries). Scores are averaged over the 10 example tasks in Table 4.

| Task | PT, 40 Queries | PT+TED, 40 Queries | PT, 70 Queries | PT+TED, 70 Queries |
|---|---|---|---|---|
| box-close-v2 | 0.10±0.04 | 0.07±0.01 | 0.06±0.02 | 0.07±0.01 |
| drawer-open-v2 | 0.18±0.06 | 0.14±0.02 | 0.21±0.06 | **0.29±0.11** |
| hammer-v2 | 0.26±0.06 | 0.57±0.19 | 0.23±0.00 | **0.49±0.07** |
| handle-pull-side-v2 | 0.25±0.27 | 0.79±0.14 | 0.01±0.00 | **0.58±0.04** |
| sweep-v2 | 0.86±0.08 | 0.64±0.30 | **0.90±0.00** | 0.53±0.13 |
| pick-place-v2 | 0.01±0.00 | 0.01±0.00 | 0.00±0.00 | 0.01±0.00 |
| plate-slide-back-v2 | **0.30±0.09** | 0.06±0.02 | 0.22±0.01 | 0.24±0.02 |
| push-v2 | 0.11±0.04 | 0.27±0.23 | 0.29±0.05 | **0.67±0.12** |
| reach-wall-v2 | 0.44±0.02 | **0.82±0.02** | 0.81±0.13 | 0.79±0.06 |
| sweep-into-v2 | 0.43±0.05 | 0.48±0.23 | **0.76±0.05** | 0.62±0.05 |
| Average | 0.29±0.07 | **0.39±0.06** | 0.34±0.03 | **0.42±0.06** |

Table 12: Ablation study on reward model quality (100 queries and ground-truth reward). Scores are averaged over the 10 example tasks in Table 4.

| Task | PT, 100 Queries | PT+TED, 100 Queries | PT, True Reward | PT+TED, True Reward |
|---|---|---|---|---|
| box-close-v2 | 0.42±0.20 | 0.42±0.02 | **0.94±0.01** | **0.93±0.01** |
| drawer-open-v2 | 0.20±0.01 | 0.59±0.02 | 0.76±0.01 | **0.84±0.07** |
| hammer-v2 | 0.43±0.07 | 0.46±0.02 | **1.22±0.06** | 0.87±0.11 |
| handle-pull-side-v2 | 0.23±0.14 | **0.43±0.03** | 0.27±0.14 | 0.26±0.14 |
| sweep-v2 | 0.84±0.09 | 0.49±0.04 | **0.88±0.00** | 0.87±0.01 |
| pick-place-v2 | 0.01±0.00 | 0.12±0.02 | **0.53±0.16** | 0.50±0.02 |
| plate-slide-back-v2 | **0.30±0.04** | 0.22±0.02 | 0.23±0.02 | 0.24±0.01 |
| push-v2 | 0.34±0.16 | 0.67±0.12 | 0.78±0.16 | **0.86±0.06** |
| reach-wall-v2 | 0.39±0.03 | 0.79±0.00 | **0.91±0.01** | **0.92±0.02** |
| sweep-into-v2 | 0.54±0.05 | 0.65±0.01 | 0.76±0.00 | **0.80±0.05** |
| Average | 0.36±0.07 | 0.44±0.02 | **0.72±0.06** | **0.71±0.05** |

Table 13: Improvement of OPRL-ATAC+TED over OPRL-ATAC.

| Task | ATAC+True Reward | OPRL-ATAC | OPRL-ATAC+TED |
|---|---|---|---|
| box-close-v2 | **0.46±0.03** | 0.12±0.01 | 0.26±0.03 |
| drawer-open-v2 | **0.95±0.07** | 0.37±0.04 | **0.83±0.09** |
| hammer-v2 | **0.26±0.01** | **0.27±0.00** | **0.28±0.01** |
| handle-pull-side-v2 | **0.77±0.23** | 0.01±0.00 | 0.27±0.18 |
| sweep-v2 | **0.79±0.17** | 0.01±0.00 | **0.73±0.12** |
| pick-place-v2 | 0.00±0.00 | 0.00±0.00 | **0.01±0.00** |
| plate-slide-back-v2 | **0.25±0.04** | 0.15±0.01 | **0.24±0.02** |
| push-v2 | **0.11±0.07** | 0.01±0.00 | 0.07±0.07 |
| reach-wall-v2 | **0.97±0.01** | 0.57±0.18 | **0.95±0.01** |
| sweep-into-v2 | **0.70±0.18** | 0.24±0.16 | **0.65±0.05** |
| Average | **0.53±0.09** | 0.17±0.04 | **0.43±0.06** |

Table 14: Performance of PT and PT+DatasetTruncating on 50 Meta-World MT1 tasks. PT+DatasetTruncating discards information of low-preference regions and underperforms PT.

| Task | True Reward | PT | PT+DataTruncating |
|---|---|---|---|
| assembly-v2 | **0.25±0.13** | 0.10±0.00 | 0.06±0.00 |
| basketball-v2 | **0.90±0.00** | 0.00±0.00 | 0.00±0.00 |
| bin-picking-v2 | **1.12±0.01** | 0.01±0.00 | 0.01±0.00 |
| box-close-v2 | **0.94±0.01** | 0.15±0.00 | 0.11±0.01 |
| button-press-topdown-v2 | **0.75±0.00** | 0.66±0.02 | 0.16±0.06 |
| button-press-v2 | **0.92±0.00** | 0.68±0.15 | 0.22±0.00 |
| coffee-button-v2 | **0.30±0.03** | 0.25±0.00 | 0.28±0.00 |
| coffee-pull-v2 | **0.54±0.04** | 0.01±0.00 | 0.01±0.00 |
| coffee-push-v2 | **0.81±0.04** | 0.02±0.00 | 0.10±0.10 |
| disassemble-v2 | **0.53±0.01** | 0.08±0.01 | 0.08±0.01 |
| door-close-v2 | **0.80±0.00** | 0.68±0.02 | **0.81±0.01** |
| door-lock-v2 | **0.78±0.07** | **0.74±0.02** | 0.15±0.00 |
| door-open-v2 | **0.58±0.05** | 0.24±0.03 | 0.07±0.03 |
| door-unlock-v2 | **0.84±0.02** | **0.86±0.01** | 0.77±0.01 |
| drawer-close-v2 | 0.53±0.00 | 0.52±0.01 | **0.74±0.01** |
| drawer-open-v2 | **0.76±0.01** | 0.13±0.00 | 0.10±0.00 |
| faucet-close-v2 | 0.77±0.04 | 0.79±0.03 | **0.88±0.04** |
| faucet-open-v2 | 0.78±0.01 | **0.82±0.03** | 0.38±0.07 |
| hammer-v2 | **1.22±0.06** | 0.23±0.00 | 0.23±0.00 |
| hand-insert-v2 | **0.80±0.01** | 0.01±0.00 | 0.00±0.00 |
| handle-press-side-v2 | **0.57±0.30** | 0.24±0.00 | 0.04±0.00 |
| handle-press-v2 | **0.97±0.00** | 0.30±0.00 | 0.28±0.01 |
| handle-pull-side-v2 | **0.27±0.14** | 0.03±0.00 | 0.01±0.00 |
| handle-pull-v2 | **0.29±0.18** | 0.03±0.00 | 0.01±0.00 |
| lever-pull-v2 | **0.41±0.04** | 0.28±0.01 | 0.12±0.01 |
| peg-insert-side-v2 | **0.79±0.01** | 0.08±0.03 | 0.00±0.00 |
| peg-unplug-side-v2 | **0.37±0.10** | 0.25±0.05 | 0.14±0.03 |
| pick-out-of-hole-v2 | **1.23±0.15** | 0.35±0.16 | 0.00±0.00 |
| pick-place-v2 | **0.53±0.16** | 0.01±0.00 | 0.00±0.00 |
| pick-place-wall-v2 | **0.04±0.01** | 0.00±0.00 | 0.00±0.00 |
| plate-slide-back-side-v2 | **0.36±0.27** | 0.15±0.00 | 0.13±0.01 |
| plate-slide-back-v2 | **0.23±0.02** | **0.20±0.03** | 0.13±0.01 |
| plate-slide-side-v2 | **0.30±0.03** | 0.24±0.01 | 0.09±0.07 |
| plate-slide-v2 | **0.86±0.01** | 0.12±0.00 | 0.08±0.01 |
| push-back-v2 | **1.15±0.05** | 0.01±0.00 | 0.00±0.00 |
| push-v2 | **0.78±0.16** | 0.03±0.02 | 0.00±0.00 |
| push-wall-v2 | **0.97±0.20** | 0.35±0.37 | 0.01±0.00 |
| reach-v2 | **0.95±0.00** | 0.81±0.05 | 0.46±0.24 |
| reach-wall-v2 | **0.91±0.01** | 0.85±0.02 | 0.46±0.02 |
| soccer-v2 | **0.42±0.17** | **0.39±0.14** | 0.25±0.07 |
| stick-push-v2 | **0.33±0.22** | 0.01±0.00 | 0.00±0.00 |
| sweep-v2 | **0.88±0.00** | 0.18±0.16 | 0.04±0.00 |
| sweep-into-v2 | **0.76±0.00** | 0.10±0.00 | 0.04±0.02 |
| window-close-v2 | 0.47±0.07 | **0.52±0.04** | 0.13±0.00 |
| window-open-v2 | **0.43±0.00** | 0.19±0.05 | 0.05±0.01 |
| button-press-topdown-wall-v2 | 0.18±0.09 | 0.05±0.10 | **0.47±0.22** |
| dial-turn-v2 | **0.70±0.04** | 0.15±0.02 | 0.01±0.00 |
| button-press-wall-v2 | **0.75±0.02** | 0.68±0.07 | 0.63±0.02 |
| shelf-place-v2 | **0.28±0.29** | 0.00±0.00 | 0.00±0.00 |
| stick-pull-v2 | **0.18±0.01** | 0.03±0.03 | 0.01±0.00 |
| Average | **0.65±0.07** | 0.27±0.03 | 0.18±0.02 |

Table 15: Comparison between IQL with true reward, IQL with random reward, and IQL with reward model learned with OPRL and 10 queries. OPRL-I achieves true-reward level performance with merely 10 queries.

| D4RL Task | True Reward | Random Reward | 10 Queries |
|---|---|---|---|
| hopper-random-v2 | 0.08±0.00 | 0.01±0.00 | **0.14±0.01** |
| hopper-medium-expert-v2 | **0.98±0.09** | 0.71±0.03 | **1.09±0.01** |
| hopper-medium-replay-v2 | 0.63±0.16 | 0.49±0.03 | **0.95±0.02** |
| hopper-medium-v2 | **0.54±0.02** | 0.53±0.00 | 0.55±0.01 |
| hopper-expert-v2 | **1.10±0.00** | **1.10±0.00** | **1.10±0.00** |
| halfcheetah-random-v2 | **0.03±0.00** | 0.02±0.00 | 0.02±0.00 |
| halfcheetah-medium-expert-v2 | **0.85±0.03** | 0.64±0.02 | 0.70±0.05 |
| halfcheetah-medium-replay-v2 | 0.35±0.01 | **0.37±0.01** | 0.36±0.00 |
| halfcheetah-medium-v2 | **0.43±0.00** | 0.42±0.00 | 0.42±0.00 |
| halfcheetah-expert-v2 | **0.93±0.00** | **0.93±0.00** | **0.93±0.00** |
| ant-random-v2 | 0.29±0.01 | **0.31±0.00** | **0.31±0.00** |
| ant-medium-expert-v2 | **1.27±0.01** | 1.21±0.02 | 1.16±0.02 |
| ant-medium-replay-v2 | **0.73±0.02** | 0.57±0.03 | 0.72±0.02 |
| ant-medium-v2 | **0.90±0.02** | **0.90±0.02** | **0.90±0.01** |
| ant-expert-v2 | **1.28±0.02** | 1.26±0.02 | 1.26±0.03 |
| walker2d-random-v2 | **0.05±0.00** | **0.06±0.01** | 0.00±0.00 |
| walker2d-medium-expert-v2 | **1.09±0.00** | **1.09±0.00** | **1.09±0.00** |
| walker2d-medium-replay-v2 | **0.75±0.07** | 0.63±0.11 | 0.74±0.03 |
| walker2d-medium-v2 | **0.80±0.01** | 0.78±0.02 | 0.75±0.02 |
| walker2d-expert-v2 | **1.08±0.00** | **1.08±0.00** | **1.08±0.00** |
| antmaze-medium-diverse-v2 | **0.68±0.08** | 0.27±0.14 | **0.82±0.03** |
| antmaze-medium-play-v2 | 0.71±0.03 | 0.20±0.05 | **0.80±0.04** |
| antmaze-large-diverse-v2 | **0.43±0.03** | 0.05±0.02 | **0.39±0.06** |
| antmaze-large-play-v2 | 0.43±0.05 | 0.04±0.02 | **0.53±0.08** |
| antmaze-umaze-diverse-v2 | **0.68±0.02** | 0.27±0.09 | 0.32±0.20 |
| antmaze-umaze-v2 | **0.93±0.01** | 0.87±0.02 | 0.88±0.05 |
| Average | **0.69±0.03** | 0.57±0.03 | **0.69±0.03** |

