# OpenReview forum: "Efficient Offline Preference-Based Reinforcement Learning with Transition-Dependent Discounting"
_ICLR.cc/2024/Conference — ICLR 2024 Conference Withdrawn Submission_

### Official Review · Reviewer_AQAk · 2023-10-30

**Soundness:** 2 fair
**Presentation:** 2 fair
**Contribution:** 2 fair
**Rating:** 1
**Confidence:** 4

**Summary:**

This paper considers the offline preference-based reinforcement learning setting, which learns a reward function and a policy from a set of trajectories and pairwise preference label among the trajectories. In particular, it considers an scenario in which the reward model is imperfect due to the lack of data. The key idea is to rank trajectories by their predicted returns and down-rate transitions that belong to low-return trajectories in TD learning. The authors verified the efficacy of this idea with experiments on a variety of manipulation tasks.

**Strengths:**

1. The imperfection of learned reward models is indeed an important topic for PbRL.
2. The proposed method is verified on a variety of manipulation tasks.

**Weaknesses:**

1. The assumption that learned reward models can identify low-preference regions more precisely than high-preference regions is quite strong but lacks justification and verification. More in-depth investigations, especially quantitative ones, are necessary here. For example, this is affected by the way preference queries are generated, which is not considered in this paper.

2. Label scarcity is an issue for both online PbRL and offline PbRL, and semi-supervised learning is a natural fit for this setting. For example, a solution is already proposed in [1]. Yet this paper does not discuss or offer results for semi-supervised learning approach.

3. The paper does not consider the quality of labels, i.e. label noise, which is an apparent issue as the proposed method leverages estimated returns for ranking trajectories.


[1] SURF Semi-supervised Reward Learning with Data Augmentation for Feedback-efficient Preference-based Reinforcement Learning

**Questions:**

1. Could you provide justification and evidence for the assumption mentioned in Weakness #1?
2. How does this method perform on the human-generated dataset provided by PT?

---

### Official Review · Reviewer_CqDq · 2023-10-30

**Soundness:** 1 poor
**Presentation:** 2 fair
**Contribution:** 3 good
**Rating:** 3
**Confidence:** 4

**Summary:**

The authors present a method for reducing the amount of preference data required for offline preference-based reinforcement learning. The method is based on reward models being good at detecting low-quality regions of the problem space and uses this ability to scale down the discount factor in low-quality regions of the problem space. By scaling down the discount factor the authors keep the learned q-values small and keep the policy from exploring into low-valued regions. The authors additionally introduce a MetaWorld based benchmark in place of the standard D4RL benchmark due to the ability of policies to learn on D4RL with poor quality rewards making it poorly suited for offline preference-based reinforcement learning. The authors evaluate their method (TED) in conjunction with two offline algorithm and compare against a baseline trained on the true reward and another offline PbRL method. Across methods, TED improves the quality of learned policies.

**Strengths:**

- The paper is well written and easy to read.
- The inclusion of transition-dependent discounting improves policy performance relative to the baseline methods. For PT, the increase in performance is particularly large.
- Point out an issue with D4RL for offline PbRL and introduce a new benchmark in its place that does not suffer from the same limitations (i.e. pessimistic policies can solve the tasks even with poor quality reward functions).

**Weaknesses:**

- The authors motivate the work with the observation in Figure 1 the that the reward model is good at detecting low quality data. However, in Figure 1, especially for pick-place-v2, the low-value regions according to the ground truth returns cover the whole span of predicted returns. It is the relative ordering and differences between regions that are important to be able to predict, not the absolute ground truth reward value. The values in peg-unplug-size-v2 look like they support the authors' claim a little more, but overall Figure 1 undermines the authors' claim that the model is good at detecting low quality data.
- The motivating plot (Figure 1) should normalize both axes to the same range. It shouldn't be expected that the learned reward function would recover the absolute values of the ground truth reward function. Instead it is the relative value of states and the relative difference in reward between states that is important. The figure seems to focus on confirming absolute value prediction.
- There are no comparisons to other methods for predicting transition/state specific discount factors.
- Table 4 would be easier to parse if all results were reported relative to the "True Reward" result and then the best performing algorithm was bolded.
- It would be helpful to plot PT with ground truth reward and PT+TED with ground truth reward as horizontal lines in Figure 3.
- It would be helpful to list the scores in the "Offline Algorithm" results subsection like you do for "Reward Model Quality" and "Truncating the Dataset"

**Questions:**

- Why use the top-k% of trajectories in order of the bottom-k% for the discount model in equation 5, since the motivating claim is that the low-preference regions are easy for the model to predict?
- Does the "True Reward" condition use TED? Does the method provide benefit to policy learning when the reward function is known?
- Do you have a hypothesis for why PT has a much larger performance gain when TED is incorporated compared to OPRL?
- In the results sections, when you talk about "normalized scores" are these the task success rates?
- How many random seeds are the experiments run over?
- For Figure 3, how do the other algorithms compare with different amounts of feedback?
- How did you pick the 10 tasks you chose to highlight in the results section?

---

### Official Review · Reviewer_KeDu · 2023-11-01

**Soundness:** 2 fair
**Presentation:** 3 good
**Contribution:** 2 fair
**Rating:** 3
**Confidence:** 4

**Summary:**

The paper first identifies an intriguing phenomenon where, in the preference learning, the approximation of reward model in low-preference regions is more accurate compared than high-preference regions. The paper introduces Transition-dependent Discounting (TED), which assigns lower discount factors to low-preference regions. To further advance research on preference learning in offline scenarios, the paper also introduces a more challenging dataset.

**Strengths:**

1. The discovered phenomenon is highly valuable and meaningful, as it has the potential to advance the learning of more accurate reward models in preference learning.
2. The proposed method is also straightforward, effective, and widely applicable.

**Weaknesses:**

1. The major shortcoming of the paper lies in the disconnection between the observed phenomenon and the proposed method. Given that an inaccurate reward model can lead to inaccurate predictions in high-preference regions, methods to address this phenomenon should be proposed. However, the proposed approach appears to weaken the impact on accurately predicting low-preference regions. Although Transition-dependent Discounting (TED) is effective and reasonable for downstream tasks, it seems unrelated to the phenomenon observed in the paper.
2. The proposed method lacks innovation and seems more like an application of this paper *On the Role of Discount Factor in Offline Reinforcement Learning* in preference learning.

**Questions:**

1. The presentation of Figure 1 is not clear. We suggest adding the $y=x$ line to the graph. Values around this line are estimated accurately, while those deviating from it are less accurate.
2. How are the top-k trajectories selected? If they are chosen based on predictions from the reward model, considering the previously observed phenomenon where high-preference regions have significant errors, would this affect the final results?

---

### Official Review · Reviewer_wPHc · 2023-11-01

**Soundness:** 2 fair
**Presentation:** 1 poor
**Contribution:** 2 fair
**Rating:** 3
**Confidence:** 4

**Summary:**

The paper introduces a transition-wise discounting factor that penalizes trajectories that are less-preferred in general. The proposed method outperforms the considered baselines in Meta-World MT1 tasks.

**Strengths:**

- The proposed method is simple yet does improve sample efficiency and performance compared to the baselines.
- The paper provides in-depth analysis on why D4RL, a popular benchmark in offline RL, should be avoided for offline PbRL.

**Weaknesses:**

- In the introduction section, the definition of the term "low preference" is ambiguous. Does it refer to trajectory segments with low predicted returns? Then, the term "preference" should not be used as the preference for a trajectory segment in PbRL generally is defined only by comparing to other trajectory segments. Noting some segments are "low preference" in absolute fashion is a little ackward.
- The presentation of Figure 1 could be improved. It is hard to checkout the density of the datapoints. I recommend reducing the markersize or applying some alpha.
- It is unclear why the authors chose to use a specific technique (transition-wise discount factors) to account for the generalization error, as there could be many alternative approaches for applying penalties. To name a few, CQL [1] directly penalizes the Q-values and TD3+BC [2] simply adds a BC term.
- Equation (2) has some typos (unclosed parentheses).
- The connection between the observation and the proposed method is somewhat unclear. Why is the proposed transition-wise discount factor required when you have better prediction accuracy on segments with low predicted return?
- How does the method work when the Q-values are negative (or contain negative values) ? According to Equation (5), it seems the proposed method will bias the learned policy **towards** the low-preference regions.
- Why is the discount factor determined trajectory-wise, when you have transition-wise reward predictions? Does the proposed method underperform when the discount factor is determined transition-wise? If so, it seems the actual benefit of the proposed method comes from smoothing the learned reward model, as is observed similarly from [3].
- It is unclear what Figure 2 is trying to convey. First, how is the plot produced in detail (how are each dot drawn) ? What does Average Q (y-axis) refer to? Also, it is hard to determine whether Average Q vs. Ground-Truth Return correlation is actually improved in (c) compared to (b). Can the authors provide a quantitative metric to show this?
- The experiments were not conducted on real human preference, which limits the scope of the work.
- Typo on Table 3: ML1 -> MT1 ?
- Figure 3. notes the proposed method "does not harm performance if ground-truth rewards are used", but the figure does not contain any information about ground-truth rewards.
- On Table 2, what reward (or return) information does Top 10% BC use? Is it ground-truth reward?
- On Section 4.1, the authors note "The remaining tasks that TED does not have notable improvement in are either too hard to learn even with true rewards or are so easy that they can be solved with baseline algorithms.", but from Table 4 there exist tasks that true reward columns have high score but the baselines and the proposed methods have low scores (e.g., box-close-v2, hammer-v2).
- More information on Table 5 would be nice. How does the correlation occur on *-medium-v2, *-medium-replay-v2, hopper, and antmaze? Also, could the authors specify exactly which dimension does the high correlation occur? It would be meaningful since in D4RL tasks, each dimension has specific (interpretable) meanings.
- Figure 4 seems to imply that in D4RL, the segments with low predicted returns actually have a larger prediction error. This result goes against the paper's initial observation in Figure 1
- It would be informative to provide Ground-truth returns vs Predicted return plot for all the tasks considered. (which will mean an extension of Figure 1 and Figure 4). This will help determine if the lower predicted return regions do actually have large prediction errors.

[1] Kumar et al., Conservative Q-Learning for Offline Reinforcement Learning, NeurIPS 2020.

[2] Fujimoto et al., A Minimalist Approach to Offline Reinforcement Learning, NeurIPS 2021.

[3] An et al., Direct Preference-based Policy Optimization without Reward Modeling, NeurIPS 2023.

**Questions:**

Mentioned above.